# High-Resolution Computed Tomography as an Initial Diagnostic and Localization Tool in Patients with Cerebrospinal Fluid Rhinorrhea: A Meta-Analysis

**DOI:** 10.3390/medicina59030540

**Published:** 2023-03-10

**Authors:** Do Hyun Kim, Sung Won Kim, Jae Sang Han, Geun-Jeon Kim, Mohammed Abdullah Basurrah, Se Hwan Hwang

**Affiliations:** 1Department of Otolaryngology-Head and Neck Surgery, Seoul Saint Mary’s Hospital, College of Medicine, The Catholic University of Korea, 222 Banpo-daero Seocho-gu, Seoul 06591, Republic of Korea; 2Department of Surgery, College of Medicine, Taif University, Taif 21944, Saudi Arabia; 3Department of Otolaryngology-Head and Neck Surgery, Bucheon Saint Mary’s Hospital, College of Medicine, The Catholic University of Korea, 222 Banpo-daero Seocho-gu, Seoul 06591, Republic of Korea

**Keywords:** cerebrospinal fluid, cerebrospinal fluid leak, outcome assessment, health care, diagnosis, meta-analysis

## Abstract

*Background and Objectives*: This study was performed to investigate the utility of high-resolution computed tomography (HRCT) for the initial localization of cerebrospinal fluid rhinorrhea. *Methods*: HRCT data regarding the point of cerebrospinal fluid leakage (as confirmed in the operating room), collected up to December 2022, were extracted from five databases. The risk of bias of the included studies was assessed using the Quality Assessment of Diagnostic Accuracy Studies 2 tool. *Results*: The search revealed eight relevant studies with a total of 254 patients. The diagnostic odds ratio of the imaging studies was 10.0729 (95% confidence interval [CI]: 2.4486; 41.4376; I^2^ = 54.1%). The area under the summary receiver operating characteristic curve was 0.8. Sensitivity, specificity, the negative predictive value, and the positive predictive value were 0.7550 (95% CI: 0.6163; 0.8553; I^2^ = 69.8%), 0.8502 (95% CI: 0.5986; 0.9557, I^2^ = 49.3%), 0.4106 (95% CI: 0.2418; 0.6035; I^2^ = 59.0%), and 0.9575 (95% CI: 0.8955; 0.9834; I^2^ = 27.7%), respectively. *Conclusions*: HRCT can be used to accurately localize cerebrospinal fluid rhinorrhea because it shows bony defects in high detail. However, it has limited utility for the evaluation of active leakage, and localization is difficult in the presence of coexisting lesions.

## 1. Introduction

Cerebrospinal fluid (CSF) rhinorrhea can occur spontaneously, as a result of trauma, and after surgery. If the fistula persists, free communication between the nasal cavity and intracranial space can provide an open channel for the transmission of infection [1]. Meningitis occurs in 10–25% of patients with unresolved CSF leaks, 10% of whom die [2,3]. β2-Transferrin and β-trace protein measurement, along with radionuclide cisternography, are representative methods to confirm CSF rhinorrhea. Failure to precisely determine the leakage point may result in repair failure despite surgery. In addition, standard nasal endoscopy has limitations in determining the location of CSF leaks. In order to compensate for this point, intrathecal fluorescein is administered to help confirm the leak point with an endoscope. Therefore, radiological assessments play a crucial role in localizing leakage points in cases of CSF rhinorrhea. Various methods for the localization of CSF rhinorrhea are available, including high-resolution computed tomography (HRCT), cisternography with magnetic resonance imaging (MRI), computed tomography (CT), and radionuclide cisternography. HRCT may be useful as an initial diagnostic modality because it can be used to localize bony defects, can be performed rapidly, poses little risk, and is relatively inexpensive [4,5,6]. However, the diagnostic power of HRCT for CSF rhinorrhea differs among studies. In addition, there have been no reviews of the diagnostic accuracy, sensitivity, and specificity of HRCT for CSF rhinorrhea. Therefore, this meta-analysis was performed to confirm the diagnostic utility of HRCT for CSF rhinorrhea, and to discuss its clinical applications and provide information that will be useful for patients.

## 2. Materials and Methods

### 2.1. Study Protocol

This systematic review and meta-analysis was conducted in accordance with the Preferred Reporting Items Guidelines for Systematic Review and Meta-Analysis (PRISMA) [7]. The research protocol is prospectively registered in the Open Science Framework “https://osf.io/q7pxw/ (accessed on 10 August 2021)”.

### 2.2. Literature Search Strategy

Studies were identified in the PubMed, SCOPUS, Embase, Web of Science, and Cochrane Central Register of Controlled Trials databases up to December 2022. The search terms were as follows: “cerebrospinal fluid rhinorrhea”, “cerebrospinal fluid”, “CSF leak”, “CSF rhinorrhea”, “cerebrospinal fluid fistula”, “CSF fistula”, “diagnosis”, “localization”, “imaging”, and “computed tomography”. Two independent reviewers checked the summaries and titles of all articles retrieved from the databases, and excluded articles unrelated to the topic of interest. In cases where the two reviewers did not agree, a decision on study inclusion was reached through discussion with a third reviewer.

### 2.3. Study Inclusion Criteria

The study inclusion criteria were as follows: the enrolment of patients who underwent HRCT for the assessment of CSF rhinorrhea; cohort studies; a comparison of imaging findings and surgical results; and inclusion of data allowing sensitivity and specificity to be determined. Case reports and review articles, as well as reports that did not include data allowing sensitivity and specificity values to be derived, were excluded.

### 2.4. Data Curation and Risk of Bias Assessment

Study data were standardized for analysis purposes [8,9,10], and included diagnostic odds ratio (DOR), summary receiver operating characteristic (sROC) curve, and area under the curve (AUC) values. The DOR was calculated as follows: (true-positive (TP)/false-positive (FP))/(false-negative (FN)/true-negative (TN)). The DOR was used as a proxy of diagnostic accuracy, and was calculated along with 95% confidence intervals (CIs) using random-effects models that considered both within- and between-study differences [2,11,12,13,14,15,16,17]. The DOR was obtained in the context of a surgical identification of CSF leakage and ranged from 0 to infinity; higher values indicate a better diagnostic performance. A DOR of 1 indicates that the diagnostic method is of no assistance with respect to determining the presence or absence of disease, and values of 0–1 indicate an inverse correlation. The sROC curve is the most intuitive method for calculating sensitivity and specificity values for meta-analyses. As the discriminant power of the test increases, the sROC curve moves closer to the upper left corner of the ROC space, where both sensitivity and specificity are 1 (100%) [18]. The AUC has a value of 0–1, with higher values indicating greater accuracy of the diagnostic test. An AUC of 0.90–1.0 is considered to indicate excellent diagnostic accuracy, while a value of 0.80–0.90 is good, 0.70–0.80 is fair, 0.60–0.70 is poor, and 0.50–0.60 indicates diagnostic failure [19]. The Quality Assessment of Diagnostic Accuracy Studies 2 tool was used to evaluate risk of bias [20].

### 2.5. Statistical Analysis

R software (version 4.2.2; R Foundation for Statistical Computing, Vienna, Austria) was used for the meta-analysis. Heterogeneity was evaluated using the Q statistic. Forest plots of sensitivity and specificity (based on sROC curves) are presented. Egger’s test and Begg’s funnel plot test were not performed due to the small number of included studies (<10).

## 3. Results

Eight studies with a total of 254 patients were included in the analysis (Figure 1). The characteristics of the included studies and bias evaluation results are shown in Table 1 and Table 2.

The DOR of HRCT was 10.0729 (95% confidence interval [CI]: 2.4486; 41.4376; I^2^ = 54.1%) (Figure 2). The AUC was 0.8, which indicates good diagnostic accuracy (Figure 3). The sensitivity, specificity, negative predictive value, and positive predictive value were 0.7550 (95% CI: 0.6163; 0.8553; I^2^ = 69.8%), 0.8502 (95% CI: 0.5986; 0.9557, I^2^ = 49.3%), 0.4106 (95% CI: 0.2418; 0.6035; I^2^ = 59.0%), and 0.9575 (95% CI: 0.8955; 0.9834; I^2^ = 27.7%), respectively (Figure 4).

## 4. Discussion

HRCT showed a good diagnostic accuracy for CSF rhinorrhea, with a sensitivity of 75.5% and specificity of 85%. HRCT has a number of advantages: it can be performed easily and quickly, and is noninvasive and relatively inexpensive. In addition, as well as skull base bone dehiscence, overall facial bone damage can be assessed.

MRI is useful for determining mucosal pathology, but has the disadvantages of low resolution and difficulty in identifying bony structures. Therefore, a method using HRCT and MRI simultaneously was proposed [21]; however, there is an opinion that it is not effective in terms of cost-effectiveness [6]. Therefore, HRCT was preferred as a standalone diagnostic modality in several studies [4,5,6]. CT and radionuclide cisternography are advantageous for identifying leakage points. However, these methods are invasive because they require the intrathecal administration of a contrast agent through lumbar puncture, and the risk of morbidity associated with the contrast agent must be taken into consideration [22,23]. In other words, these methods have limitations as initial diagnostic tools. Magnetic resonance cisternography has the advantage of requiring no intrathecal contrast agent injection, and T2-weighted images with high signal intensity can be helpful for localizing CSF leakage points [24]. However, the lack of information on bony structures and the high cost (more than five times higher than HRCT) reduce the attractiveness of magnetic resonance cisternography for initial leakage point localization [6]. Recently, in addition to imaging tools, studies have been reported that can effectively check the CSF leak site using an endoscope equipped with a blue light filter after intrathecal fluorescein administration, increasing the number of options for diagnostic testing [24].

Taking the above points into account, HRCT has advantages as an initial diagnostic tool. However, the sensitivity of HRCT (75.5%) is disadvantageous in terms of its use as a screening tool. CT can provide indirect information about CSF fistula formation, but cannot be used to directly confirm leakage [25]. In addition, basilar defects of <2 mm are difficult to discriminate using HRCT. If the bone is thin due to mucocele, meningocele, or sinusitis, or if bone signals overlap, it may be difficult to determine the leakage point. Moreover, even in cases with multiple skull base fractures, there are limits to the accuracy of leakage point localization. As the sensitivity is low, a diagnosis should also be based on clinical symptoms and confirmatory tests [25,26].

Our meta-analysis had several limitations. First, differences in the quality of imaging equipment and imaging techniques among institutions may have affected diagnostic accuracy. Second, differences in image reading proficiency among clinicians may have affected the results. Therefore, care should be taken when interpreting the clinical results. Finally, most studies did not describe the etiology of CSF rhinorrhea, and did not determine whether the leaks were active or inactive.

## 5. Conclusions

This meta-analysis showed that HRCT is useful for localizing CSF rhinorrhea. However, given its limitations, HRCT should be combined with other tests for diagnostic confirmation as necessary.

## Figures and Tables

**Figure 1 medicina-59-00540-f001:**
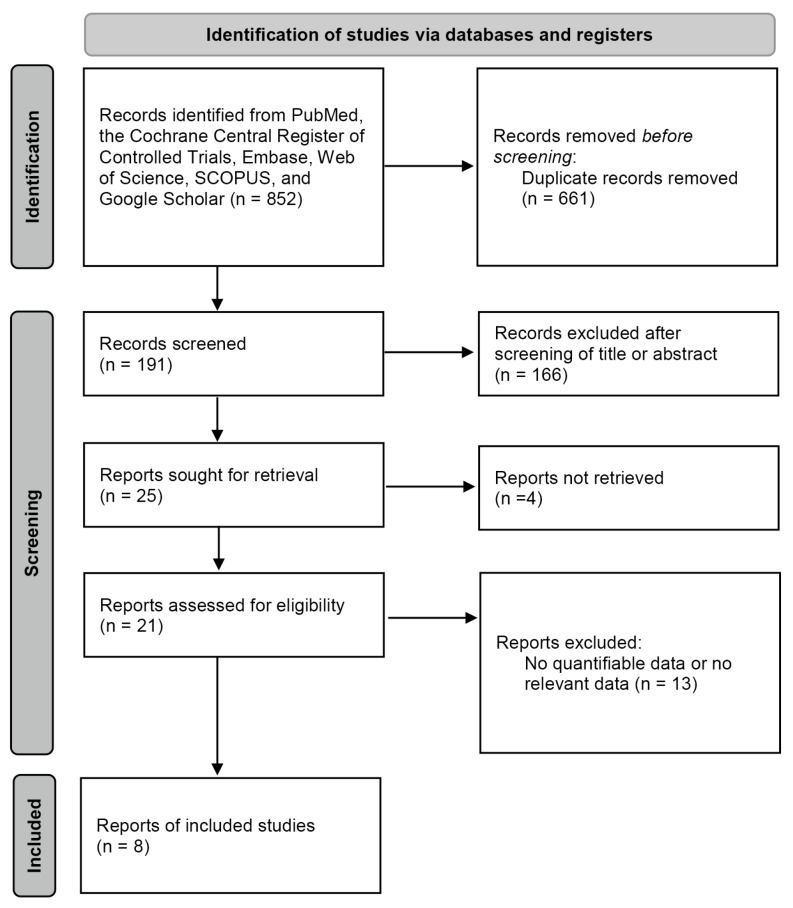
Flow diagram of the selection of studies.

**Figure 2 medicina-59-00540-f002:**
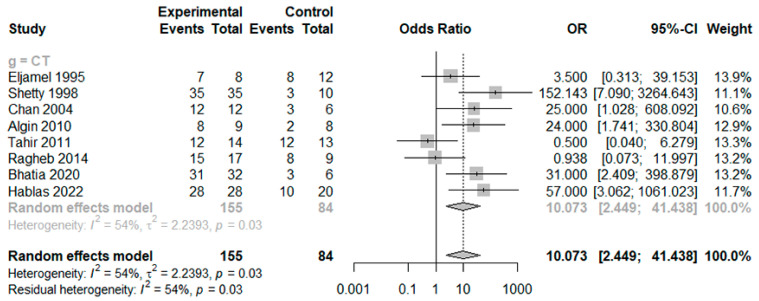
Diagnostic accuracy of included studies [2,11,12,13,14,15,16,17].

**Figure 3 medicina-59-00540-f003:**
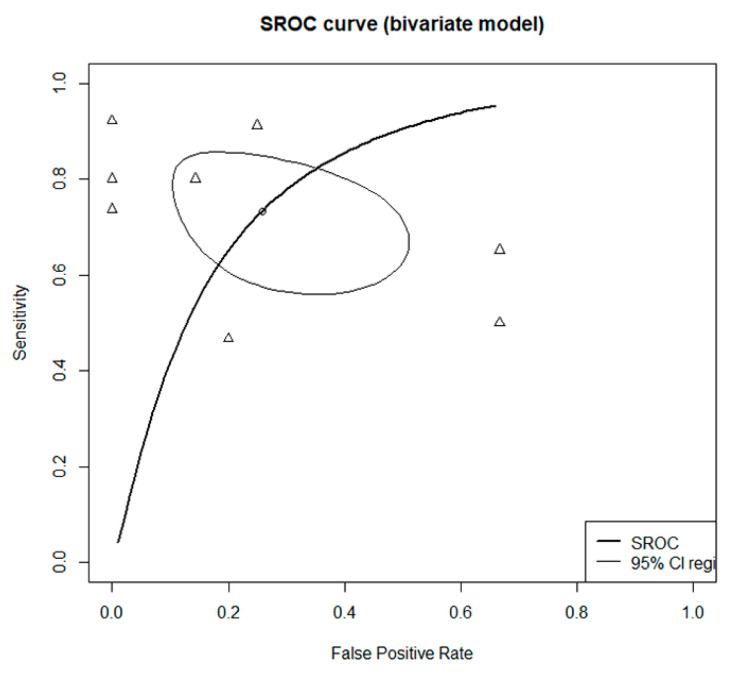
The summary receiver operating characteristic curve. Thick curved line (summary receiver operating characteristic curve), thin circular line (95% confidence region), and small circle (summary estimate).

**Figure 4 medicina-59-00540-f004:**
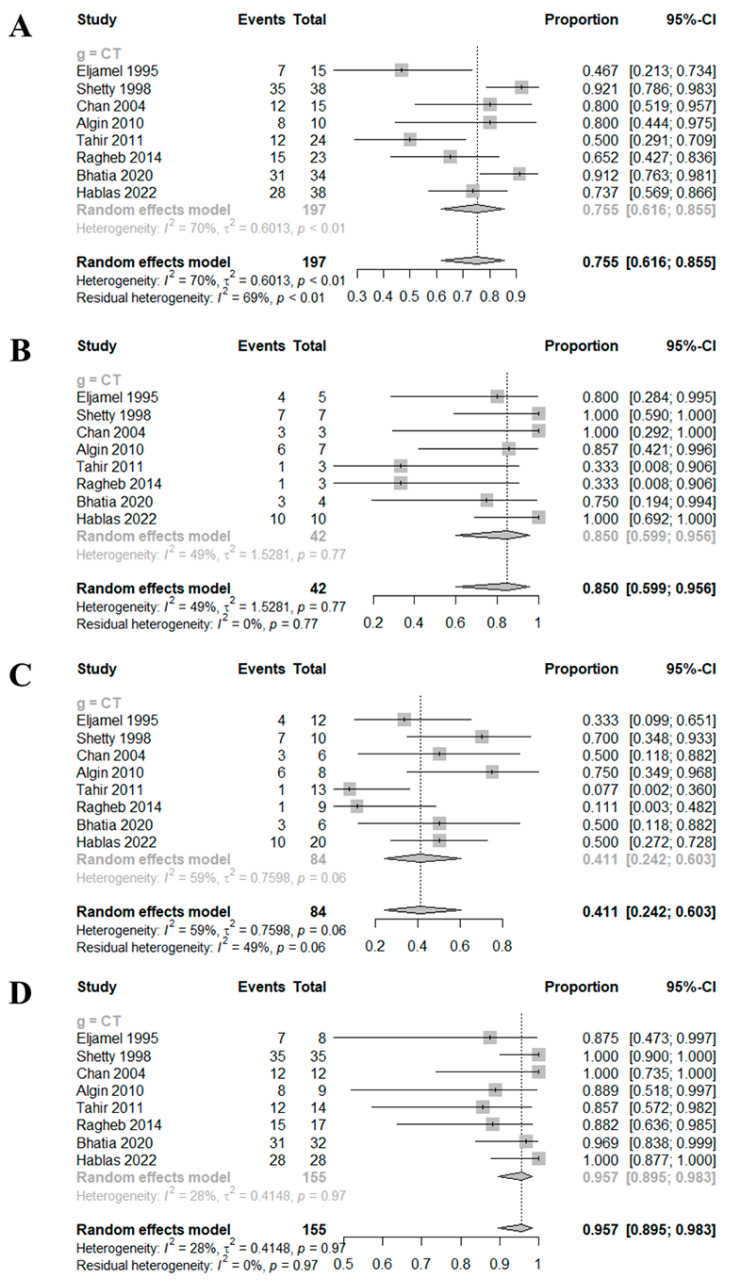
Forest plots of sensitivity (**A**), specificity (**B**), negative predictive value (**C**) and positive predictive value (**D**) [2,11,12,13,14,15,16,17].

**Table 1 medicina-59-00540-t001:** Characteristics of the included studies.

Study	Design	Number of Patients	Sex (Male/Female)	Age, Median (Range) or Mean (SD), y	Nation	Frequency of Radiologic Exam/Numbers of Lesion	TP	FN	FP	TN
Eljamel 1995 [11]	Case series	21	16/5	33 ± 14 (2–60)	Ireland	21	7	8	1	4
Shetty 1998 [2]	Cohort	45	28/17	3–62	India	45	35	3	0	7
Chan 2004 [12]	Cohort	18	NA	NA	China	18	12	3	0	3
Algin 2010 [13]	Cohort	17	13/4	32 (11–70)	Turkey	17	8	2	1	6
Tahir 2011 [15]	Cohort	43	17/26	40.6 (3–74)	Pakistan	43	12	12	2	1
Ragheb 2014 [14]	Cohort	24	16/8	33–62	Egypt	24	15	8	2	1
Bhatia 2020 [16]	Cohort	38	21/17	NR	India	38	31	3	1	3
Hablas 2022 [17]	Case series	48	20/28	19–67	Egypt	48	28	10	0	10

Abbreviations: NR, not reported; TP, true positive; TN, true negative; FP, false positive; FN, false negative.

**Table 2 medicina-59-00540-t002:** Methodological quality of the included studies.

Reference	Risk of Bias				Concerns about Application		
	Patient Selection	Index Test	Reference Standard	Flow and Timing	Patient Selection	Index Test	Reference Standard
Eljamel 1995 [11]	Low	Low	Unclear	Low	Low	Low	Low
Shetty 1998 [2]	Low	Low	Low	Low	Low	Low	Low
Chan 2004 [12]	Low	Low	Unclear	Low	Low	Low	Low
Algin 2010 [13]	Low	Low	Unclear	Low	Low	Low	Low
Tahir 2011 [15]	Unclear	Low	Unclear	Unclear	Low	Low	Low
Ragheb 2014 [14]	Low	Low	Unclear	Low	Low	Low	Low
Bhatia 2020 [16]	Low	Low	Low	Low	Low	Low	Low
Hablas 2022 [17]	Low	Low	Low	Low	Low	Low	Low

## Data Availability

The raw data of individual articles used in this meta-analysis are included in the main text.

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
