# Peer review of "High-Resolution Computed Tomography as an Initial Diagnostic and Localization Tool in Patients with Cerebrospinal Fluid Rhinorrhea: A Meta-Analysis"

_medicina, 2023, doi:10.3390/medicina59030540_

Round 1

Reviewer 1 Report

Intersting topic with proper research of studies included.

It is correctly stated that HRCT is only an initial diagnostic tool with a sensitivity of 75% which requires other diagnostic tools to prove the correct location of CSF leakage.

Please mention lumbar fluorescein injection for blue light endoscopy detection in the nose as an alternative in diagnostics (ll. 42-44 and 148-149).

Author Response

Reviewer 1

Please mention lumbar fluorescein injection for blue light endoscopy detection in the nose as an alternative in diagnostics (ll. 42-44 and 148-149).

 Reply:

We added information in the introduction and discussion section that intrathecal fluorescein administration can be helpful in identifying CSF leak points. A mention of the blue light filter has been added to the discussion.

Reviewer 2 Report

This is well-prepared meta-analysis showing the importance of HRCT to detect the CSF rhinorrhea via bone defects. The results were presented with well prepared figures. The discussed well.

The authors investigated the importance of HRCT to detect the CSF rhinorrhea via bone defects.

It is interesting, because CSF rhinorrhea is important problem. Detecting the bone defect is important to diagnose. This is an easy method. Other techniques for analyzing CSF or rhinorrhea liquid for beta-transferrin is more difficult

Not too original, but important and interesting and helper of the other methods

Detecting the bone defect by HRCT  is an easy method. Other techniques for analyzing CSF or rhinorrhea liquid for beta-transferrin is more difficult

 Paper is well written, the text is clear and easy to read, the conclusions are consistent with the evidence and arguments presented. They address the main question posed?

Author Response

Reviewer 2

Paper is well written, the text is clear and easy to read, the conclusions are consistent with the evidence and arguments presented. They address the main question posed.

 Reply:

Thank you for your positive comments.

Reviewer 3 Report

The authors conducted a rigorous analysis of CSF leak via a meta-analysis. The findings are valid and clinically relevant.

The data were well presented in the forms of Figures and supplementary tables. As characteristics of studies( Table S1) and ROB (Table S2) are essential to the interpretation of the findings, I would like to suggest these two tables to be presented in the main document instead of as supplementary data.

Author Response

Reviewer 3

The data were well presented in the forms of Figures and supplementary tables. As characteristics of studies( Table S1) and ROB (Table S2) are essential to the interpretation of the findings, I would like to suggest these two tables to be presented in the main document instead of as supplementary data.

 Reply:

We moved Table S1 and S2 to Table 1 and 2.